# The Onset of Menstruation and Social Networking Site Use in Adolescent Girls: The Mediating Role of Body Mass Index

**DOI:** 10.3390/ijerph18199942

**Published:** 2021-09-22

**Authors:** Chenyu Lv, Ofir Turel, Qinghua He

**Affiliations:** 1Faculty of Psychology, Southwest University, Chongqing 400715, China; lvchenyu96@163.com; 2Faculty of Engineering and Information Technology, University of Melbourne, Parkville, Victoria 3053, Australia; oturel@unimelb.edu.au; 3Key Laboratory of Cognition and Personality (Ministry of Education), Southwest University, Chongqing 400715, China; 4Chongqing Collaborative Innovation Center for Brain Science, Chongqing 400715, China; 5Collaborative Innovation Center of Assessment toward Basic Education Quality, Beijing Normal University, Chongqing 400715, China

**Keywords:** social media, social networking sites, menstruation, BMI, menarche, adolescent girls

## Abstract

Evidence suggests that girls spend much time on social networking sites (SNS), often more than boys do. It has been proposed that this may have to do with sex-based differences in the need and approaches for socialization. We posit that adolescent girls are also unique in that they are developing physiologically and start menstruating. Based on prior research, we hypothesize that the onset of menstruation can drive physiological changes (increased body mass index (BMI)), which together with common behavioral–psychological (social and emotional) responses to menstruation can contribute to an increase in SNS use. We therefore aim to test whether BMI partially mediates the relationship between menstruation and SNS use in adolescent girls. Results based on a large nationally representative sample in the United Kingdom suggest that the age of menarche was negatively associated with daily hours of SNS use, and that BMI elevation partially mediated this association. These results extend the negative effects of the early onset of menstruation and imply that BMI control strategies may help to reduce the use of SNS in girls who experience menarche at an earlier age.

## 1. Introduction

### 1.1. The Use of Social Networking Sites

Social networking sites (SNS) are Internet-based platforms that combine messaging, photo albums, journals, media management and other services [1]. They provide adolescents with an effective way to interact with each other and build relationships beyond their proximal geography [2]. As such, in many cases they partially (and sometimes fully) replace or subtitle traditional face-to-face communications [3]. They are highly popular. For instance, Facebook has more than 1 billion monthly active users [4]. More than 50% of adolescents use it [5]. A key reason for such trends in adolescents is the fact that social networking sites have rich functions that can allow adolescents to meet their evolving social needs, express themselves, create desirable images of themselves and shelter themselves from the more vulnerable true-self [6,7]. Consequently, SNS afford the avoidance of reality, dealing with social anxiety and portraying a more desirable self, and these affordances drive social networking site use in adolescents [8], and especially in girls [9]. For instance, in 2013–2016 the time adolescent girls spent on SNS was higher than that of adolescent boys [10]. One British study interviewed nearly 11,000 14-year-olds and found that many girls use social media much longer than boys. Two in five girls spent at least three hours a day on social media, compared with just one in five boys of the same age [11]. While normal use of SNS is not harmful, excessive use can be and can result in adverse social, behavioral, psychological and physiological effects that infringe upon normal functioning in the family, school and social life domains [12,13,14]. It has been argued that the widespread adoption of SNS by adolescents between 2009 and 2012 has contributed to a decrease in adolescent well-being in the English-speaking world [15,16]. Thus, it is important to study what drives SNS use. Findings of such studies can serve as a basis for interventions for people who present excessive use patterns and for detecting risk factors for excessive use. Thus, we aim to explain SNS use in adolescent girls from a unique angle.

### 1.2. The Physical and Mental Effects of Menarche

To do so, we first note that social anxiety in adolescents may be due to the physical development that is common in this stage of life. Social anxiety development processes are especially pronounced in girls [17], and one reason for this is that they start menstruation [18]. Menstruation is the periodic discharge of blood and linings of the uterine mucosa through the vagina. The onset of this discharge is called menarche; it begins at or before sexual maturity and stops at menopause [19]. The age of menarche is an important indicator of the maturation of the female reproductive system, which shows the effectiveness of the female reproductive hormones, the follicle stimulating hormone and luteinizing hormone [20]. The age of menarche matters because the early onset of menstruation can typically lead to undesirable physiological and psychological effects [21,22]. An earlier onset of menstruation can create long lasting changes in estrogen levels in females [23], changes in body structure and social interactions, increased tendency to engage in rebellious behavior [24], elevated obesity [25] and even elevated offspring obesity [26].

Studies have shown that girls who have already experienced menarche had significantly higher weight, bone, muscle, body fat and body fullness compared to those of the same age who did not experience menarche; the earlier the menarche occurred, the more significant the difference between the two groups was [27]. At this time, girls not only experience physiological changes, but also undergo many psychological changes, which together can affect their body image and responses to social pressures [28]. Particularly, an early age of menarche increases one’s risk for psychosocial distress, eating disorders, earlier sexual initiation, substance abuse and behavioral addictions [29]. One psychological reason is that girls at menarche may experience feelings of shame, fear, anxiety and depression, leading to more negative emotional reactions [30]. In addition, after menarche, girls change in role recognition and self-consciousness, and can feel dissatisfied with home and school environments [31]. Moreover, early menarche indicates to girls that they differ from their peers, and this can exacerbate socialization concerns [32]. Researchers used the Pubertal Developmental Scale (PDS) to survey 2430 adolescents and found that early puberty was positively related to adolescent Internet use, both in boys and girls [33]. Together, such findings indicate that girls who experience menarche at an early age may be more likely than others to prefer interactions over social media than face-to-face, because the former affords presenting a more idealized version of themselves and a lower sense of shame compared to face-to-face socialization [34]; SNS communication also affords escaping the often harsh post-menarche social reality, such as feelings of inferiority and unpleasantness brought on by physical changes [35]. Taken together, in the present study, it is hypothesized that adolescent girls with earlier menarche are associated with more frequent use of social networking sites (Hypothesis 1) and that the level of BMI of adolescent girls is negatively associated with the age of menarche (Hypothesis 2).

### 1.3. Social Networking Sites and BMI

Studies have shown a link between BMI and Internet use. For example, there is an association between Internet addiction (IA) and BMI in healthy adolescents, including in middle school students [36,37]. There is also a bidirectional association between technology use (or screen time) and obesity: screen time can lead to obesity, but also obesity can lead to increased screen time [38]. Thus, it is possible that overweight individuals avoid face-to-face social interactions to some extent and favor sedentary over-the-Internet interactions, including via SNS. It is also conceivable that obese people feel increased shame, depressive symptoms and social exclusion [39,40], and hence prefer social interactions in a sheltered environment, in which they can portray themselves anyway they want, and certainly in a more positive way [41,42]. Indeed, overweight children were more stigmatized among their peers, which can hurt the self-esteem of overweight children and may result in low motivation toward social interaction [43,44]. Therefore, social isolation in obese people may result in the individual spending more time on SNS [45]. As such, it is hypothesized that BMI will be positively related to an elevated use of SNS (Hypothesis 3). This means that the relationship between age at menarche and the use of SNS is at least partially mediated by BMI (Hypothesis 4).

### 1.4. The Present Study

Against this theoretical and empirical background, data from a large sample were analyzed in order to investigate whether BMI and the age of menarche were associated with the use of social networking sites in adolescent girls (Hypotheses 1–4). Although these hypotheses have to some extent been tested in previous research, most empirical studies have relied upon small, targeted samples of girls or neglected the role of physiological changes (e.g., menarche, BMI) in social networking site use. Consequently, the present study contributes to the literature in at least two important ways. First, the large sample of adolescent girls that was recruited increases the generalizability of the findings. Second, we have a physiological explanation for the use of SNS among adolescent girls, making it a novel and specific addition to this research field.

## 2. Materials and Methods

### 2.1. Procedure

Data were extracted from the age 14 sweep (first sweep that includes females who started menstruating) of the UK Millennium Cohort Study (MCS). See a full description of the study in Joshi and Fitzsimons [46]. In a nutshell, these data were collected in 2015–2016 by the Centre for Longitudinal Studies, London, UK. The MCS follows longitudinally, in sweeps conducted every several years, a sample of 18,818 babies born in 2000–2002 in England, Scotland, Wales and Northern Ireland. Due to attrition, the age 14 sweep contained 11,884 adolescents [47]. The survey is collected via a phone interview with guardians and adolescents. Participation required consent from children and assent from guardians. The procedures were approved by relevant ethics committees. The MCS collects hundreds of variables pertaining to many aspects of life. Given the focus of this study, we extracted only adolescent girls from the full dataset, and only responses pertaining to our variables of interest.

### 2.2. Measures

Participants were asked “On a normal weekday during term time, how many hours do you spend on social networking or messaging sites or Apps on the internet such as Facebook, Twitter and WhatsApp?” The question used the following 1–8 Likert-type scale: 1 = None, 2 = Less than half an hour, 3 = Half an hour to less than 1 h, 4 = 1 h to less than 2 h, 5 = 2 h to less than 3 h, 6 = 3 h to less than 5 h, 7 = 5 h to less than 7 h, 8 = 7 h or more. Information regarding menarche was collected through a self-reported questionnaire. To capture the onset of menstruation (age at menarche), participants were asked the questions: “Have you started your periods?” and “How old were you when you had your first period?” BMI was extracted from self-obtained physical measurements of weight and height.

### 2.3. Statistical Analysis

SPSS 26.0 and the PROCESS macro [48] were used for all analyses. Participants with missing data for the independent, mediating or outcome variables were excluded from analyses.

## 3. Results

### 3.1. Participant Characteristics

Focusing on only girls in the sample has left us with an initial sample of 5931 adolescents, ages 13–15 (M = 13.77; SD = 0.45). Out of them, 5462 answered the menstruation question, and 5119 reported they started menstruation (90.7%). Out of them, 4737 responded to the SNS use question and agreed to provide physical measures pertaining to BMI. This has resulted in an operational sample of 4734 adolescent girls (79.8% of initial sample; Age range = 13–15, M_age_ = 13.79; SD_age_ = 0.44). The average age of menarche among adolescents was 12.16 years old, with the earliest being 7 years old and the latest 15 years old. The characteristics of the sample are presented in Table 1 (mean ± SD for continuous variables). Descriptive statistics for the three body size measures—body weight, height and the body mass index (computed as follows: BMI = weight/height2)—are presented along with age, onset age of menstruation and hours per day spent on social networking sites (SNS).

### 3.2. Correlation

Correlations among the focal variables are presented in Table 2. BMI was negatively associated with the onset of menstruation. This means the younger the onset age of menstruation was, the higher the BMI was. Menstruation onset was also negatively associated with the use time (hours/day) of social networking sites. This means that girls with earlier menstruation onset used social networking sites more often than those with later menstruation onset. Additionally, the association between BMI and SNS use time was significantly positive. The higher the BMI was, the more time adolescent girls spent on their social networking sites.

#### 3.2.1. Menstruation and BMI

The group that experienced menarche (*n* = 5119) was compared to those who did not (*n* = 523) with independent sample *t*-tests. The results showed that the BMI of the menstruating group was significantly higher than that of the non-menstruating group, *t* (5334) = 14.889, *p* < 0.000. This indicates that BMI and weight are related to the occurrence of menarche. In addition, we explored the relationship between the level of BMI and the age of menarche. BMI was divided into three groups according to the 2007 World Health Organization (WHO) standards for girls by age, namely the normal group (17.2 ≤ BMI ≤ 22.7), the low body mass index group (BMI < 17.2) and the high body mass index group (BMI > 22.7). The difference in menarche age among the three groups was in the expected direction. The results showed that the age of menarche in the high BMI group was significantly lower than that in the other groups, F = 88.769, *p* < 0.000.

#### 3.2.2. The Occurrence of Menarche and SNS Use

Independent sample *t*-test results showed that hours per day spent on SNS in the menstruation group were significantly higher than those of the non-menstruating group, *t* (5637) = 11.629, *p* < 0.000. This indicates that SNS use is positively associated with the occurrence of menarche.

#### 3.2.3. BMI and SNS

We examined the use of social networking sites at different BMI levels (normal group, low body mass index group and high body mass index group). The results of univariate analysis of variance indicated that the social networking site use time of groups with different levels of BMI was significantly different, F = 27.807, *p* < 0.000. Post hoc multiple comparisons pointed to differences between the low body mass index group (Mean = 4.30, SD = 1.97) and the normal group (Mean = 5.03, SD = 1.97), and between the low body mass index group and the high body mass index group (Mean = 5.10, SD = 2.04). These differences mean that adolescent girls with a higher BMI spent more hours on SNS per day. However, there was no significant difference between the normal and the high BMI groups.

### 3.3. Mediation Analysis

Our theoretical account predicts that BMI at least partially mediates the negative effect of menstruation onset on SNS use. We tested these hypotheses with bootstrapping with 5000 re-samples using Model #4 in the PROCESS macro for SPSS. The results are depicted in Figure 1. Both the direct effect [−0.171; −0.059] and the mediated through BMI effect [−0.024; −0.001] had 95% non-standardized confidence intervals that exclude zero. These results indicated that BMI partially mediated the negative relationship between menstruation and social networking site use.

## 4. Discussion

In this study, we provide a unique physiological, and indirectly psychological, explanation for SNS use among adolescent girls. We posited that this can depend, in part, on the age of onset of menstruation and the resultant BMI increases, both of which can contribute to elevated levels of SNS use. We therefore explored the association between age at menarche and SNS use and examined the mediational effect of BMI on this association. We found that early menarche was significantly associated with a longer use of SNS per day. Moreover, results from the mediation analysis confirmed the hypothesis that the relationship between age at menarche and SNS use was partially mediated by BMI increases.

Our study showed that age at menarche was negatively associated with SNS use time. The hours per day spent on SNS decreased along with an increase in age at menarche, and late menarche was significantly related to less time spent on SNS. Although no study has directly demonstrated a relationship between age at menarche and SNS use, previous studies have shown that girls who experienced menarche can face many psychological changes, including the emergence of negative emotions such as shame, fear, anxiety or depression, which may lead to problematic behaviors, including online behaviors [29,49]. Similarly, menarche indicates changes in sexual maturity, which can also affect one’s need for and forms of socialization [50,51,52]. Such psychological changes may be partly responsible for the increased use of SNS among adolescent girls because SNS afford sheltered, frequent and easier to manipulate social interactions compared to face-to-face environments. Hence, we open the door for future research to examine the aforementioned psychological mechanism, which we did not capture in the current study.

However, another part of the explanation may relate to physiology, and more specifically to changes in BMI imposed by menarche. Menarche is the onset of puberty and therefore brings physical changes, most notably an increase in height or weight. Studies have demonstrated the effect of menarche on the physical development and body type of girls, which is consistent with our findings in this study. The BMI of girls who experienced menarche was significantly higher than that of same age girls who did not start menstruating. Moreover, the earlier the participants started menstruating, the higher their BMI was, which is consistent with previous research [27]. Furthermore, the relationship between BMI and SNS use has been confirmed by previous studies [36,37,38]. Adolescence is an important period for the physical and psychological development of young people. An increase in BMI can lead to significant changes in body shape, leading to possible ostracism, isolation, social interaction avoidance and to increased willingness to spend more time socializing over the Internet [45]. In this study, we also found that BMI was significantly positively correlated with the use time of SNS. The higher the BMI of adolescent girls was, the more time they spent on SNS. In addition, we observed significant differences in SNS use time between the different BMI groups. Similar results have been found in children. Studies have shown that overweight children tend to protect themselves from negative comments and attitudes by staying in a safe place such as their homes and spending more time in sedentary activities, which may lead to unhealthy weight gain [53].

An interesting observation we make here is that the association between age at menarche and SNS use is partially mediated by BMI. This means that the age of menarche, by affecting BMI, can eventually lead to changes in the duration of SNS use. This can explain, at least in part, why adolescent girls tend to use SNS more than adolescent boys do [10]. However, it should be noted that there may be many confounds not identified in this study that can affect SNS use and/or BMI. Hence, further studies are needed to increase the generalizability of our findings. From a practical standpoint, our findings provide one reasonable explanation for SNS use time. While normally SNS use time is not an issue, if therapists or physicians want to help adolescent girls to reduce SNS use time (e.g., in cases of excessive use, cyberbullying or excessive sedentary time), our findings suggest that being aware of the age at menarche is relevant, and that strategies for curbing high BMI can be helpful. In addition, just being aware of the association between age at menarche, BMI and SNS use may help adolescents avoid the risks of excessive use of SNS, sedentary time and increased food intake. Considering these are sensitive topics, we can make brochures, early sex education readers or social networking manuals, or even some relevant public service advertisements. Such approaches and assertions, though, merit further research.

This is the first study to explore the association between age at menarche and SNS use among adolescent girls in the UK. Its strengths include a large sample size and a large collection of relevant health information and physiological measures. We not only explored the relationship between age at menarche and SNS use time, but also studied a mechanism through which age at menarche influences SNS use time. Nevertheless, this study has several potential limitations that should be considered. Firstly, the age at menarche, the height and the weight were self-reported by participants, which may have recall bias. However, the self-report method is acceptable to a certain extent [54]. Secondly, this was a cross-sectional study that could not prove causal relationships. Future studies can use longitudinal designs or experiments to better establish causality. Thirdly, we focus on a limited set of factors, as dictated by our theory. Nevertheless, there are other factors that may influence the use of SNS, such as loneliness and depression among adolescents, which were not taken into account in the model, or other factors that can affect BMI, such as genetics, social economic status and eating habits. Future research can expand our model to account for such factors. Finally, we only discussed one form of Internet use in this study—social networking sites—while other types of Internet use, such as online games [55,56,57,58] or shopping [59], may produce different results. Prior research has compared problematic gaming and problematic social media use and found that the two problematic uses have slightly different associations with psychological health and sleep [60]. Future research could further explore the relationship among different types of Internet use, BMI and menarche.

## 5. Conclusions

Adolescent girls with early menarche are likely to spend more time on SNS compared to others. This is likely due to both psychological and physiological changes imposed by menarche. We found that BMI is a relevant physiological change that mediates the relationship between age at menarche and SNS use time. We call for more research into all aspects of the physical and psychological effects of adolescents’ use of technology such as social media sites, smartphones and video games.

## Figures and Tables

**Figure 1 ijerph-18-09942-f001:**
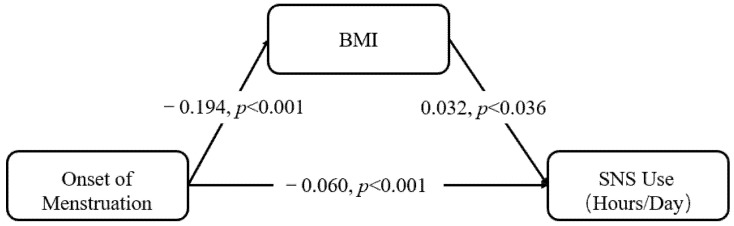
Mediation analysis of the effect of menstruation on SNS use through BMI. Note: Standardized coefficient and *p*-values are reported for each path.

**Table 1 ijerph-18-09942-t001:** Descriptive statistics (Operational *n* = 4734).

Characteristic	Mean ± SD	Minimum
Height (cm)	161.42 ± 6.17	139.5
Weight (kg)	58.02 ± 11.68	20
Body Mass Index (BMI)	22.22 ± 4.05	7.35
Age (year)	13.79 ± 0.44	13
Age of start of menstruation	12.16 ± 1.03	7
Social networking site (SNS) use	5.12 ± 1.98	1
Height (cm)	161.42 ± 6.17	139.5

Notes: SD is standard deviation. BMI = weight/height2. (Daily) social networking site use was measured on a 1–8 scale. A mean of 5.12 represents between 2 h and less than 3 h per day, or 2.12 h per day, if linearly extrapolated.

**Table 2 ijerph-18-09942-t002:** Associations between BMI, menstruation onset age and SNS use.

	BMI	Menstruation	SNS Use
BMI	1		
Menstruation onset	−0.194 **	1	
SNS use	0.044 **	−0.066 **	1

** *p* < 0.01.

## Data Availability

The data presented in this study are available on request from the corresponding author. The data are not publicly available due to privacy. The full data may be extracted from the Millennial Cohort Study site upon approval by the data owners.

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
