# Peer review of "The Onset of Menstruation and Social Networking Site Use in Adolescent Girls: The Mediating Role of Body Mass Index"

_ijerph, 2021, doi:10.3390/ijerph18199942_

Round 1

Reviewer 1 Report

Overall, I found the paper interesting, the writing clear, and the analysis well-focused and appropriately linked to the hypotheses identified. 

The paper draws attention to the significance and influence of gender and developmental milestones in relation to adolescent girls' social media use.

Health and wellbeing concerns in relation to girls' social media use would be useful information to include in the introduction (would strengthen significance). For example, empirical evidence (stats) about wellbeing issues experienced by them in relation to their social media use.

Line 84: The points raised about 'desired self' and 'free of shame' need to be tempered and/or explained better (e.g., consider tone).

Line 85: 'harsh post-menarche social reality' needs to be explained in plain language and then the idea linked to your hypotheses.

The two points above are better explained in lines 225-235 - perhaps draw on those latter explanations to clarify these points earlier in the paper.

Line 222: 'longer use' needs to be clarified (e.g., time online in one sitting or time from first starting to use SNS).

Line 245: 'previous studies' ... see for example ... ?

Line 268-270:  These are touchy subjects to discuss with adolescent girls - often taboo - so how do you work around these social concerns while helping them to avoid excessive use of SNS?

Line 298: This final sentence is not clear.

Minor spelling and punctuation errors need correction.

Author Response

Point 1: Health and wellbeing concerns in relation to girls' social media use would be useful information to include in the introduction (would strengthen significance). For example, empirical evidence (stats) about wellbeing issues experienced by them in relation to their social media use.

Response 1: Thanks for the advice, and it has been added in the manuscript from line 48-50. ‘Researchers have been claimed that the widespread adoption of SNS by adolescents between 2009 and 2012 has contributed to a decrease in adolescent well-being in the English-speaking world since 2012 [15,16]’.

Point 2: Line 84: The points raised about 'desired self' and 'free of shame' need to be tempered and/or explained better (e.g., consider tone).

Response 2: It has been revised in the manuscript from line 86-87. ‘a more idealized version of themselves’ and ‘a lower sense of shame’.

Point 3: Line 85: 'harsh post-menarche social reality' needs to be explained in plain language and then the idea linked to your hypotheses.

Response 3: 'harsh post-menarche social reality' has been further explained in the manuscript, ‘it also affords escaping the often harsh post-menarche social reality, such as feelings of inferiority and unpleasantness brought on by physical changes.’

The two points above are better explained in lines 225-235 - perhaps draw on those latter explanations to clarify these points earlier in the paper.

Point 4: Line 222: 'longer use' needs to be clarified (e.g., time online in one sitting or time from first starting to use SNS).

Response 4: 'longer use' means ‘spend more time on SNS per day’. It has been revised in the manuscript from line 222-223. ‘We found that early menarche was significantly associated with a longer use of SNS per day.’

Point 5: Line 245: 'previous studies' ... see for example ...?

Response 5: We added related references here in line 246.

Point 6: Line 268-270:  These are touchy subjects to discuss with adolescent girls - often taboo - so how do you work around these social concerns while helping them to avoid excessive use of SNS?

Response 6:In this study, we explored the relationship among age of menarche, BMI and time of SNS use. The findings suggested that there are ways that adolescent girls can avoid risks such as excessive use of SNS. Considering these are sensitive topics, we think we can make brochures, early sex education readers or social networking manuals, or even some relevant public service advertisements. These methods should play a certain role in the influence and education of adolescent girls. We added relevant content to the discussion session.

Point 7: Line 298: This final sentence is not clear.

Response 7: Sorry for the unclear statement. It has been revised. ‘We call for more research into all aspects of the physical and psychological effects of adolescents' use of technology such as social media sites, smartphones and video games.’

Point 8: Minor spelling and punctuation errors need correction.

Response 8: Minor spelling and punctuation errors have been corrected.

Reviewer 2 Report

Title

  • I have some concerns about the psychological part included in the title, as any evaluation was performed, it is only an assumption.

Introduction

  • This session is well written and easy to follow, providing the main ideas to understand the background of the study

Methods

  • How was BMI calculated? The data was collected by asking or measuring every girl with the same apparatus?
  • Statistical analysis – it is missing information about the test that it was used and if homogeneity was tested

Results

Line 157 – please consider changing “is” to “was”

Line 157 – this information is not the same as presented in line 143. I also would suggest removing that information from the methods session, as it was a result.

Discussion

  • I do not understand how psychological evaluation was included in this study as only BMI, age of menarche and time spent in SNS was included
  • Please consider using the SNS abbreviation along the discussion session
  • Are those girls from the same economic status and had the same lifestyle? For instance, some of those girls play sports or could have a different diet? Maybe those variables could influence both BMI and time of menarche, explaining why BMI partially mediate the use of SNS

Best regards,

Ana Filipa Silva

Author Response

Point 1: I have some concerns about the psychological part included in the title, as any evaluation was performed, it is only an assumption.

Response 1: Thanks for the advice. We changed the title into ‘The onset of menstruation on social network use in adolescent girls: the mediation role of BMI’.

Point 2: How was BMI calculated? The data was collected by asking or measuring every girl with the same apparatus?

Response 2: Sorry for the unclear statement. BMI was calculated from self-reported height and weight. This does have some drawbacks. We have added statement about how BMI was calculated and explained this as a limitation in the discussion section.

Point 3: Statistical analysis – it is missing information about the test that it was used and if homogeneity was tested

Response 3: Sorry about the missing information. The detail was mainly presented in 2.2 Measures.

Point 4: Line 157 – please consider changing “is” to “was”

Response 4: ‘is’ has been changed to ‘was’ in the manuscript line 158.

Point 5: Line 157 – this information is not the same as presented in line 143. I also would suggest removing that information from the methods session, as it was a result.

Response 5: Sorry for the inconsistency of the information. It has been revised in the manuscript and removed that information from the methods session.

Point 6: I do not understand how psychological evaluation was included in this study as only BMI, age of menarche and time spent in SNS was included.

Response 6: We are sorry for not being clear. The title has been revised. In this study, we only measured the BMI, age of menstruation and time spent in SNS of the participants, but these indicators all contain certain psychological reasons and are not pure physiological indicators. The psychological reasons behind these indicators were also explained to some extent in the introduction and discussion session in the manuscript.

Point 7: Please consider using the SNS abbreviation along the discussion session.

Response 7: Thanks for the advice. We revised the discussion session by using the SNS abbreviation.

Point 8: Are those girls from the same economic status and had the same lifestyle? For instance, some of those girls play sports or could have a different diet? Maybe those variables could influence both BMI and time of menarche, explaining why BMI partially mediate the use of SNS.

Response 8: In this study, we also investigated the family weekly income of adolescent girls, and further analysis found that it was significantly correlated with BMI, SNS and age of menarche (Table 1), indicating that economic status may indeed have a certain influence on them.

Table 1 Correlation among family weekly income, BMI, SNS and age of menarche

BMI

age of menarche

SNS use

Family weekly income

-0.134**

0.070**

-0.079

In addition, as for living and eating habits, some of the data were collected, such as the daily intake of vegetables and fruits, and the intake of sweet drinks, etc. It turns out that some were significantly correlated and some were not (Table 2). Suggesting that some living and eating habits may also affect both BMI and time of menarche. However, these measurements were relatively crude and were not the focus of this study. This content has been added in the discussion session from line 291-292 as the limitation of this study.

Table 2 Correlation among living and eating habits, BMI, SNS and age of menarche

BMI

age of menarche

SNS use

Days per week eats breakfast

-0.160**

0.084**

-0.241**

How often do you drink sweetened drinks

-0.030*

0.022

-0.217**

How often do you eat fast food

-0.020

0.041**

-0.278**

How often do you eat at least 2 portions of fruit

-0.031*

0.031*

-0.199**

How often do you eat at least 2 portions of vegetables

-0.045**

0.022

-0.155**